# Association between socioeconomic status and cardiovascular disease by sex: Mediating roles of psychological and behavioral factors

Jiwon Choi[1], Sung-il Cho[1,2]*

1 Department of Public Health Science, Graduate School of Public Health, Seoul National University, Seoul, South Korea, 2 Institute of Health and Environment, Seoul National University, Seoul, South Korea

* persontime@hotmail.com

## Abstract

Previous research shows that low socioeconomic status (SES) increases the risk of cardiovascular disease (CVD) and contributes to health disparities through the unequal distribution of intermediary factors. This study aims to examine mediation effects of psychological and health behavior factors in the SES–CVD pathway. Also, given the limitations of using a single SES indicator, we aimed to address these gaps by employing latent class analysis to construct a composite measure of SES. Our study analyzed data from the Korea Health Panel Survey, collected between 2009 and 2018. A total of 11,265 participants aged 30 and above, with no prior diagnosis of CVD and no missing responses, were included in the study. SES was derived using latent class analysis based on four variables: income, education, working status, and health insurance, conducted separately by sex. Causal mediation analysis was used to examine the pathway between SES and CVD by sex. Three SES classes were identified separately for men and women. Among men, low SES accounted for 12.7% (n = 639), medium SES for 63.1% (n = 3,177), and high SES for 24.2% (n = 1,218); among women, low SES accounted for 17.3% (n = 1,075), medium SES for 71.8% (n = 4,471), and high SES for 11.0% (n = 685). Among women, low SES was associated with a 18% shorter average survival time until CVD compared to medium SES. This association was partially mediated by depressed mood, accounting for 6.7% of the total effect, and by perceived anxiety/depression, accounting for 8.2%. Our findings indicate that psychological factors partially mediate the association between low SES and CVD among women, highlighting sex-specific pathways in socioeconomic health disparities and underscoring the importance of incorporating mental health considerations into cardiovascular prevention strategies.

**Data availability statement:** Data cannot be shared publicly due to the request of the data-providing institution. Data are available from the Korea Institute for Health and Social Affairs (KIHASA) (contact via khp@kihasa.re.kr), and researchers interested in the data may download them from the KIHASA website (https://www.khp.re.kr:444/web/data/board/dataDownload.do?bbsid=107).

**Funding:** The author(s) received no specific funding for this work.

**Competing interests:** The authors have declared that no competing interests exist.

## Introduction

Cardiovascular disease (CVD) is a leading cause of death worldwide. In 2020, CVD was responsible for approximately 19.05 million deaths globally [1]. In South Korea, CVD was the leading cause of death until 1999 and has since remained the second most common cause, continuing to be a major contributor to mortality [2]. Also, the years of life lost due to circulatory system diseases are estimated at 824 per 100,000 population [3]. Furthermore, the disability-adjusted life-year at 3,226 per 100,000 population show a continuous increase over time, suggesting a steady rise in the overall CVD burden in Korea.

CVD incidence and mortality are influenced by multiple cardiovascular risk factors (CVRFs), such as smoking, alcohol consumption, and obesity [4]. Social, environmental, and economic factors also play a role, and socioeconomic status (SES) has been identified as a significant predictor of CVD and its associated risk factors [5]. Individuals with low SES exhibit higher CVD incidence rates, likely due to the greater prevalence of CVRFs [6]. Conceptual frameworks propose that differences in social position influence health outcomes through intermediary determinants [7]. This framework is also applicable to CVD, with societal conditions influencing outcomes through psychological, behavioral, and biological pathways [8].

Low income is associated with having depression and anxiety, both of which represent established independent risk factors for CVD [9,10]. Meta-analyses have shown that low income is associated with increased depression [11]. Individuals with depression may experience dysregulation of the sympathetic nervous system and the hypothalamic–pituitary–adrenal axis, leading to coronary vasoconstriction, endothelial dysfunction, and increased platelet activation, ultimately elevating cardiac risk [10]. Structured diagnostic interviews suggest that lower SES is associated with a higher likelihood of anxiety [12]. Stress and anxiety can promote atherosclerosis and may serve as acute triggers of major cardiac events, thereby increasing CVD risk [13,14].

Health behaviors are recognized as key mediating mechanisms linking distal structural factors to individual health outcomes [15]. Health behaviors are shaped by multifaceted social, economic, and environmental factors and they exhibit strong social patterning [16]. Smoking is a major risk factor for CVD, and individuals who smoke have a significantly higher risk of CVD compared to non-smokers [17]. Sedentary behavior and physical activity are modifiable risk factors for CVD, with sedentary behavior increasing the risk of CVD and higher levels of physical activity reducing the risk [18]. These results can be explained by physical activity improving lipid profiles—raising high-density lipoprotein and lowering triglycerides and total cholesterol—which in turn reduces CVD risk.

Previous studies have explored psychological and behavioral mediators in the SES–CVD association. Depression was reported to account for 10.9% of this relationship, although the cross-sectional design limited causal inference [19]. Cigarette smoking and physical inactivity explained part of the association between social determinants of health (SDoH) and CVD mortality, but reliance on a composite SDoH score obscured the specific contributions of individual attributes [20]. Other research has identified smoking, physical inactivity, alcohol use, and body mass index (BMI)

as significant mediators of the association between SES and ischemic heart disease mortality in both men and women; however, the proportions mediated differed by sex [21].

The association between SES and CVD, as well as the mediating factors, appears to differ between men and women. Low SES has been linked to higher CVD risk, with this trend more pronounced in women [22,23]. This is because women with lower SES tend to exhibit a worse profile of cardiovascular biomarkers [24]. Moreover, the influence of SES on factors such as diet, physical activity, and psychosocial aspects may also differ by sex [22]. Importantly, the pathways through which SES influences CVD may differ in strength between men and women. For example, smoking substantially contributes to socioeconomic inequalities in ischemic heart disease mortality, accounting for a larger proportion among men (29%) than women (16%) [21]. In contrast, psychosocial factors—particularly depression and stress—together with behavioral and lifestyle influences disproportionately affect women, with depression approximately twice as prevalent in women as in men, which increases their risk of CVD [25]. Therefore, these observations underscore the need for sex-specific mediation analyses to better understand the underlying pathways.

In epidemiological studies, SES is typically assessed using education, income, and occupation or employment status, which is also confirmed in CVD research [26]. However, relying on a single element of SES fails to capture variation and may obscure significant social gradients in health [27]. Therefore, several researchers have suggested using multiple variables as indicators of SES [28]. However, including more than one SES indicator in regression models may violate the collinearity assumptions due to their high correlation [29]. To address this limitation, composite indices integrating multiple SES indicators have been proposed.

Exploring how to combine multiple SES measures meaningfully is important, and latent class analysis (LCA) has emerged as a useful method. LCA is a person-centered approach that categorizes populations into mutually exclusive groups based on observable indicators [30]. It enables the meaningful classification of SES using relevant measures [29]. Research using LCA to select proxy variables for composite aspects of SES is increasing [31,32]. Several studies have derived three to five SES groups using between three and nineteen measures, showing that diverse SES indicators effectively capture socioeconomic characteristics.

Despite previous research, important gaps remain. First, most studies use single variables to represent individual SES, capturing only part of its multidimensional nature [26]. However, constructing a comprehensive measure that reflects various aspects of SES is essential. Second, although theoretical evidence suggests that psychological and health behavior factors mediate the SES–CVD association, causal examinations of these pathways using longitudinal data remain limited, particularly with respect to sex differences. Therefore, our study aimed to quantify the extent to which psychological and health behavior factors mediate the association between SES and CVD by sex.

## Materials and methods

### Data and study population

This retrospective cohort study utilized the Korea Health Panel Survey (KHPS) data (version 1.7.3.), which was conducted annually from 2008 to 2018 by the Korea Institute for Health and Social Affairs and the National Health Insurance Service (NHIS) consortium. This panel employs a multi-stage stratified probability sampling method using 90% of the 2005 Population and Housing Census data across sixteen districts nationwide to maintain national representativeness. KHPS collected variables through face-to-face interviews using the computer assisted personal interviewing method [33]. Although the data were self-reported, their reliability was enhanced by the panel households' completion of a one-year health diary on medical utilization and the collection of medical receipts and year-end tax settlement records. Since information on health behaviors was collected during the second phase of the survey in 2009, this study utilized data from that year onward. Data were accessed on January 24, 2024.

From 20,395 individuals with at least one recorded entry between 2009 and 2013, a moving index was applied during the entry period to account for each individual's enrollment date. The study population was selected after applying the

following exclusions: 430 individuals with a history of CVD before the index date; 7,335 individuals younger than 30 years at baseline; 126 individuals whose mediator factors were measured before the baseline assessment or after the date of CVD onset; and 1,239 individuals who did not complete or respond to surveys on health behaviors, income, and depressed mood (**Fig 1**). Consequently, a total of 11,265 participants were included in the study.

### Variables

**Exposure variables.**  The exposure variable was SES classified as high, medium, and low, based on LCA using four variables—self-reported household income, education level, working status, and health insurance—conducted separately for men and women. Household income was classified into three groups, with the 1st quintile categorized as low, the 2nd to 4th quintiles as middle, and the 5th quintile as high. Education was classified into three levels: below elementary, middle or high school, and college or higher. Working status was categorized as none, full-time (regular employees), and part-time (temporary/daily workers). Health insurance was classified based on the following types of coverage: none (including national merit special benefits or foreign nationals not enrolled), medical aid (types 1 and 2), and NHIS. The exposure variable was measured once at baseline upon study enrollment.

**Outcome variables.**  The main outcome was newly diagnosed CVD cases, assessed through self-reported physician diagnoses. Participants reported the physician's diagnosis name and the corresponding International Classification of Diseases 10th edition (ICD-10) code during surveys of emergency, inpatient, and outpatient service use. We defined CVD as ischemic heart disease (ICD-10: I20–I25) or stroke (ICD-10: I60–I69) [34], and the outcome was operationalized as the first occurrence of disease during follow-up, identified from annually collected data after baseline enrollment. We confirmed that the timing of measurements did not violate the temporal order of exposure variables, mediator variables, and CVD onset.

**Mediator variables.**  We hypothesized potential mediators for the causal pathway between SES and incidence of CVD, as shown in **Fig 2**. Depressed mood was defined by a "yes" response to having felt sad for more than two weeks in the past year. Perceived anxiety/depression was defined as a "yes" response to feeling very or somewhat anxious or depressed. Smoking status was categorized into three groups: never-smoker, former smoker, and current smoker. Moderate physical activity was measured by the number of days participants engaged in at least 10 minutes of activity

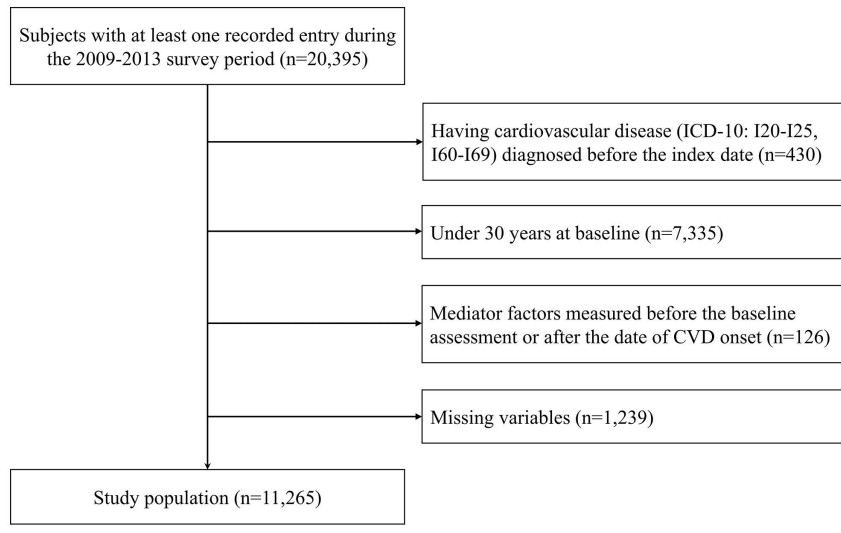

**Fig 1.  Selection of study population.**

 

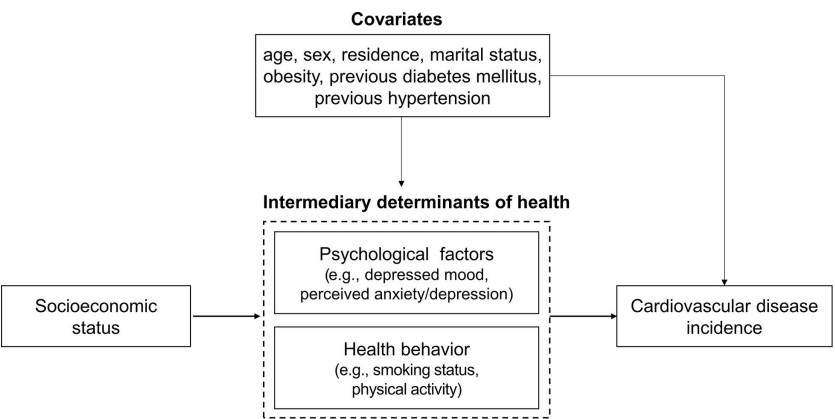

**Fig 2. The direct and indirect pathways between socioeconomic status and cardiovascular disease.** Mediator variables included psychological factors (depressed mood and perceived anxiety/depression) and health behaviors (smoking status and physical activity) in the pathway linking socioeconomic status and cardiovascular disease.

causing slightly heavier breathing, categorized as 0, 1–2, 3–5, or 6–7 days. All mediator variables were measured after the baseline assessment of the exposure variable.

**Covariates.** Based on previous research on potential confounders, the adjustments included were age, residence, marital status, obesity, history of diabetes mellitus, and history of hypertension [19,31]. Age was categorized into 30–39, 40–49, 50–59, and 60 and above, while residence was classified as Seoul metropolitan for Seoul, Gyeonggi, and Incheon and non-metropolitan for other areas [35]. Marital status was categorized as never married, married, separated/divorced, and widowed. Obesity was defined as a BMI of 25 or higher, with BMI calculated as weight divided by height squared. History of diabetes mellitus (ICD-10: E10–E14) and history of hypertension (ICD-10: I10) were defined as having the condition at baseline. However, unmeasured confounding, such as genetic factors that are difficult to measure in this study, may also exist [36].

## Statistical analysis

**Latent class analysis.** LCA is a person-centered mixture modeling approach that identifies latent subpopulations within a sample based on patterns of responses to observed categorical variables [30]. We determined the number of latent classes (ranging from two to five) based on model fit statistics and theoretical interpretability in analyses conducted separately for men and women (S1 Table). Statistical model fit was assessed using the Entropy, Akaike information criterion (AIC), Bayesian information criterion (BIC), and likelihood-ratio $G^2$ statistic. The entropy value ranges from 0 to 1, with higher values indicating better classification accuracy. AIC, BIC, and $G^2$ are goodness-of-fit statistics used to assess model fit and parsimony, where lower values indicate better-fitting models [37]. The three-class model for both men and women demonstrated the best fit based on AIC, BIC, and $G^2$ values. Its entropy values (0.63 for men and 0.69 for women) further indicate good class interpretability and separation.

**Survival analysis and model specification.** Survival analysis was performed to examine the association between SES and CVD, considering both the Cox proportional hazards model and the parametric accelerated failure time (AFT) model [38]. Log–log transformed survivor functions were evaluated to determine whether the event times across exposure groups followed a Weibull distribution and whether the hazard functions were proportional [39]. The SES-specific curves for both sexes were non-parallel with differing slopes, indicating a violation of the proportional hazards assumption (S1-S2 Figs). However, their approximately linear form supported the Weibull assumption, justifying the use of an AFT model. The

AFT model was formulated using survival functions, with the Weibull distribution showing the best fit based on AIC criteria in both men and women (S2 Table). Time was defined as the duration from baseline enrollment to the first occurrence of CVD, death, or the end of follow-up. The participants were censored by a CVD event, death, or 31 December 2018, whichever occurred first.

**Causal mediation analysis.** Causal mediation analysis provides a more general and rigorous framework for survival models, accommodating nonlinear relationships and interactions [40]. Specifically, to evaluate the causal pathways via the binary mediator, we performed mediation analysis within a counterfactual framework by integrating a logistic regression model for the mediator's conditional probability and an AFT model for the survival outcome, allowing for exposure–mediator interactions [41]. The total effect (TE) was decomposed into the natural direct effect (NDE), reflecting the exposure's effect independent of the mediator, and the natural indirect effect (NIE), representing the effect transmitted through exposure-induced changes in the binary mediator and their impact on survival under the AFT model. Causal effects were interpreted on the mean survival ratio scale, with time ratios (TRs) < 1 denoting reduced survival and TRs > 1 denoting increased survival, thereby quantifying acceleration or deceleration of event time. The proportion mediated (PM) was calculated on the mean survival ratio scale, as implemented in the SAS macro for causal mediation analysis [42]. Given that the TE in an AFT model decomposes multiplicatively into the NDE and NIE (i.e., TE = NDE × NIE), the PM was defined as [NDE × (NIE − 1)]/ (TE − 1).

**Sensitivity analysis.** For sensitivity analysis, we tested the effect modification on both the multiplicative and additive scales between SES and the binary mediators (S3 Table), in order to incorporate these findings into the causal mediation analysis. Effect modification was evaluated on the multiplicative scale using TRs derived from exponentiated regression coefficients. On the additive scale, it was assessed using the relative excess risk due to interaction (RERI), calculated by converting TRs into hazard ratios via the parameter scale and interpreting hazard ratios as an approximation of risk ratios. Confidence intervals were estimated using the delta method. Second, we conducted a sensitivity analysis to evaluate how unmeasured mediator–outcome confounding, varying in strength, prevalence, and associations, could influence the observed effects and determine the level of confounding required to alter their interpretation [43]. Combinations of confounder prevalence and confounder–mediator risk ratios ($RR_{CM}$), together with the corresponding maximum confounder–outcome risk ratios ($RR_{CY}$) that could be accommodated by the additive interaction model, were presented. As an example, we used the results of the indirect effects with depressed mood as a mediator among women and applied the additive model for the pure indirect effect (PIE) (S4 Table, S3-S8 Figs). Third, we performed analyses by substituting the composite SES derived from LCA with each SES indicators (income, education, employment status, and health insurance) to evaluate the robustness of the composite measure (S5-S8 Tables). All statistical analyses were performed using SAS V.9.4 (SAS Institute, Cary, North Carolina, USA).

## Ethics statements

This study was approved by the institutional review board of Seoul National University (IRB No. E2409/002–002). The requirement for informed consent was waived since the KHPS is anonymized data with no personal identifying information. Patients were neither involved in the recruitment nor in any other aspect of the study process. Data will be disseminated to the study participants through academic publications and public press releases.

## Results

### Characteristics of the study population

The population consisted of 11,265 individuals, with men accounting for 44.7% and women accounting for 55.3% (Table 1). Individuals aged 60 years and above accounted for the largest proportion at 31.2%. Participants with depressed mood accounted for 10.5%, and 13.3% reported perceived anxiety/depression. Never-smokers comprised the largest group for smoking status (61.5%), and individuals reporting no moderate physical activity per week constituted the highest proportion (56.9%).

 

**Table 1. Baseline characteristics of the study population (n = 11,265).**

| Variables | | N (%) |
|---|---|---|
| **Sex** | Men | 5,034 (44.7) |
| | Women | 6,231 (55.3) |
| **Age** | 30-39 | 2,649 (23.5) |
| | 40-49 | 2,814 (25.0) |
| | 50-59 | 2,285 (20.3) |
| | 60+ | 3,517 (31.2) |
| **Residence** | Seoul metropolitan | 4,644 (41.2) |
| | Non-metropolitan | 6,621 (58.8) |
| **Marital status** | Never married | 768 (6.8) |
| | Married | 9,054 (80.4) |
| | Separated/Divorced | 385 (3.4) |
| | Widowed | 1,058 (9.4) |
| **Obesity** | No | 8,429 (74.8) |
| | Yes | 2,836 (25.2) |
| **Previous diabetes mellitus** | No | 10,814 (96.0) |
| | Yes | 451 (4.0) |
| **Previous hypertension** | No | 8,485 (75.3) |
| | Yes | 2,780 (24.7) |
| **Household income** | Low (1st quintile) | 1,821 (16.2) |
| | Middle (2nd-4th quintiles) | 6,985 (62.0) |
| | High (5th quintiles) | 2,459 (21.8) |
| **Education level** | ≤ Elementary school | 2,649 (23.5) |
| | Middle-high school | 5,312 (47.2) |
| | ≥ College | 3,304 (29.3) |
| **Working status** | None | 6,955 (61.7) |
| | Part-time | 3,197 (28.4) |
| | Full-time | 1,113 (9.9) |
| **Health insurance** | None | 13 (0.1) |
| | Medical aid | 431 (3.8) |
| | NHIS | 10,821 (96.1) |
| **Depressed mood** | No | 10,083 (89.5) |
| | Yes | 1,182 (10.5) |
| **Perceived anxiety/depression** | No | 9,767 (86.7) |
| | Yes | 1,498 (13.3) |
| **Smoking status** | Never smoker | 6,932 (61.5) |
| | Former smoker | 1,737 (15.4) |
| | Current smoker | 2,596 (23.0) |
| **Moderate physical activity, times/week** | 0 | 6,410 (56.9) |
| | 1-2 | 1,249 (11.1) |
| | 3-5 | 1,998 (17.7) |
| | 6-7 | 1,608 (14.3) |

Abbreviation: NHIS, national health insurance service.

## Assessment of socioeconomic status using latent class analysis

We evaluated the characteristics of each latent class separately for men and women based on mean posterior and item-response probabilities (Table 2). In men, latent class 1 (n = 639, 12.7%) was classified as low SES, reflecting low income, low education, and non-working status. Class 2 (n = 3,177, 63.1%) represented medium SES, with intermediate levels of income and education. Class 3 (n = 1,218, 24.2%) corresponded to high SES, characterized by high income, higher education, and full-time employment. In women, latent class 1 (n = 4,471, 71.8%) represented medium SES, with intermediate income and education. Class 2 (n = 685, 11.0%) corresponded to high SES, defined by high income and

**Table 2. Mean posterior and item-response probabilities from sex-specific three-class latent class models.**

**Men**

| Item | Latent class 1 (n = 639, 12.7%) | Latent class 2 (n = 3,177, 63.1%) | Latent class 3 (n = 1,218, 24.2%) |
|---|---|---|---|
| Mean posterior probabilities | 0.85 | 0.80 | 0.94 |
| Item-response probabilities | | | |
| Income 1 (low: 1st quintile) | **0.73** | 0.05 | 0.01 |
| Income 2 (middle: 2nd-4th quintiles) | 0.25 | **0.85** | 0.45 |
| Income 3 (high: 5th quintiles) | 0.02 | 0.10 | **0.54** |
| Education 1 (<elementary) | **0.48** | 0.16 | 0.01 |
| Education 2 (middle-high school) | 0.41 | **0.63** | 0.25 |
| Education 3 (> college) | 0.11 | 0.22 | **0.74** |
| Working 1 (none) | **0.90** | **0.54** | 0.22 |
| Working 2 (part-time) | 0.10 | 0.46 | 0.24 |
| Working 3 (full-time) | 0.00 | 0.00 | **0.54** |
| Insurance 1 (none) | 0.00 | 0.00 | 0.00 |
| Insurance 2 (medical aid) | 0.20 | 0.00 | 0.00 |
| Insurance 3 (NHIS) | **0.80** | **0.99** | **1.00** |

**Women**

| Item | Latent class 1 (n = 4,471, 71.8%) | Latent class 2 (n = 685, 11.0%) | Latent class 3 (n = 1,075, 17.3%) |
|---|---|---|---|
| Mean posterior probabilities | 0.89 | 0.81 | 0.88 |
| Item-response probabilities | | | |
| Income 1 (low: 1st quintile) | 0.04 | 0.01 | **0.81** |
| Income 2 (middle: 2nd-4th quintiles) | **0.80** | 0.23 | 0.19 |
| Income 3 (high: 5th quintiles) | 0.16 | **0.76** | 0.00 |
| Education 1 (<elementary) | 0.23 | 0.00 | **0.74** |
| Education 2 (middle-high school) | **0.58** | 0.25 | 0.23 |
| Education 3 (> college) | 0.19 | **0.75** | 0.03 |
| Working 1 (none) | **0.72** | **0.48** | **0.88** |
| Working 2 (part-time) | 0.28 | 0.24 | 0.12 |
| Working 3 (full-time) | 0.00 | 0.28 | 0.00 |
| Insurance 1 (none) | 0.00 | 0.00 | 0.00 |
| Insurance 2 (medical aid) | 0.01 | 0.00 | 0.20 |
| Insurance 3 (NHIS) | **0.99** | **1.00** | **0.80** |

Note: The maximal item-response probabilities for each latent class were marked in bold. The mean posterior probabilities for all latent classes exceeded the minimum threshold of 0.8, indicating an acceptable level of classification uncertainty.

higher education. Class 3 (n = 1,075, 17.3%) was classified as low SES, reflecting low income and education. In all analyses, the reference category was set as medium SES, as it had the largest sample size and allowed for the assessment of trends across low and high SES groups.

Table 3 presents the distribution of SES obtained from sex-specific LCA. Low SES constituted the highest proportion among individuals aged 60 years or older (men: 72.0%, women: 73.7%). Smoking patterns differed by sex: current smoking was most prevalent among men, whereas never smoking was most common among women across all SES groups. The proportions of participants reporting depressed mood, perceived anxiety/depression, and no moderate physical activity per week were highest in the low SES group and higher in women.

### Mediation analysis with the accelerated failure time model

Multiplicative effects of mediators between SES and CVD showed no statistically significant associations in either men or women across all SES levels (S3 Table). On the additive scale, antagonistic effects were observed for high SES combined with depressed mood (in both men and women), perceived anxiety/depression in men, smoking status in

**Table 3. Characteristics of the study population according to sex-specific latent class analysis.**

| Variables, N (%) | | Socioeconomic status derived from latent class analysis | | | | | |
|---|---|---|---|---|---|---|---|
| | | Men | | | Women | | |
| | | Low SES | Medium SES | High SES | Low SES | Medium SES | High SES |
| **Age** | 30-39 | 34 (5.3) | 683 (21.5) | 440 (36.1) | 86 (8.0) | 1,075 (24.0) | 331 (48.3) |
| | 40-49 | 71 (11.1) | 799 (25.2) | 452 (37.1) | 83 (7.7) | 1,170 (26.2) | 239 (34.9) |
| | 50-59 | 74 (11.6) | 717 (22.6) | 248 (20.4) | 114 (10.6) | 1,042 (23.3) | 90 (13.1) |
| | 60+ | 460 (72.0) | 978 (30.8) | 78 (6.4) | 792 (73.7) | 1,184 (26.5) | 25 (3.7) |
| **Residence** | Seoul metropolitan | 162 (25.4) | 1,268 (39.9) | 629 (51.6) | 289 (26.9) | 1,885 (42.2) | 411 (60.0) |
| | Non-metropolitan | 477 (74.7) | 1,909 (60.1) | 589 (48.4) | 786 (73.1) | 2,586 (57.8) | 274 (40.0) |
| **Marital status** | Never married | 39(6.1) | 339 (10.7) | 119 (9.8) | 23 (2.1) | 165 (3.7) | 83 (12.1) |
| | Married | 509 (80.0) | 2,680 (84.4) | 1,078 (88.5) | 584 (54.3) | 3,611 (80.8) | 592 (86.4) |
| | Separated/Divorced | 55 (8.6) | 98 (3.1) | 18 (1.5) | 55 (5.1) | 155 (3.5) | 4 (0.6) |
| | Widowed | 36 (5.6) | 60 (1.9) | 3 (0.3) | 413 (38.4) | 540 (12.1) | 6 (0.9) |
| **Obesity** | No | 525 (82.2) | 2,256 (71.0) | 786 (64.5) | 796 (74.1) | 3,449 (77.1) | 617 (90.1) |
| | Yes | 114 (17.8) | 921 (29.0) | 432 (35.5) | 279 (26.0) | 1,022 (22.9) | 68 (9.9) |
| **Previous diabetes mellitus** | No | 591 (92.5) | 3,025 (95.2) | 1,178 (96.7) | 1,016 (94.5) | 4,325 (96.7) | 679 (99.1) |
| | Yes | 48 (7.5) | 152 (4.8) | 40 (3.3) | 59 (5.5) | 146 (3.3) | 6 (0.9) |
| **Previous hypertension** | No | 389 (60.9) | 2,401 (75.6) | 1,044 (85.7) | 545 (50.7) | 3,469 (77.6) | 637 (93.0) |
| | Yes | 250 (39.1) | 776 (24.4) | 174 (14.3) | 530 (49.3) | 1,002 (22.4) | 48 (7.0) |
| **Depressed mood** | No | 546 (85.4) | 2,939 (92.5) | 1,146 (94.1) | 852 (79.3) | 3,975 (88.9) | 625 (91.2) |
| | Yes | 93 (14.6) | 238 (7.5) | 72 (5.9) | 223 (20.7) | 496 (11.1) | 60 (8.8) |
| **Perceived anxiety/depression** | No | 527 (82.5) | 2,867 (90.2) | 1,125 (92.4) | 801 (74.5) | 3,855 (86.2) | 592 (86.4) |
| | Yes | 112 (17.5) | 310 (9.8) | 93 (7.6) | 274 (25.5) | 616 (13.8) | 93 (13.6) |
| **Smoking** | Never smoker | 97 (15.2) | 620 (19.5) | 306 (25.1) | 977 (90.9) | 4,263 (95.4) | 669 (97.7) |
| | Former smoker | 269 (42.1) | 981 (30.9) | 357 (29.3) | 29 (2.7) | 88 (2.0) | 13 (1.9) |
| | Current smoker | 273 (42.7) | 1,576 (49.6) | 555 (45.6) | 69 (6.4) | 120 (2.7) | 3 (0.4) |
| **Moderate physical activity, times/week** | 0 | 416 (65.1) | 1,538 (48.4) | 502 (41.2) | 785 (73.0) | 2,756 (61.6) | 413 (60.3) |
| | 1-2 | 29 (4.5) | 388 (12.2) | 297 (24.4) | 53 (4.9) | 387 (8.7) | 95 (13.9) |
| | 3-5 | 79 (12.4) | 617 (19.4) | 311 (25.5) | 116 (10.8) | 738 (16.5) | 137 (20.0) |
| | 6-7 | 115 (18.0) | 634 (20.0) | 108 (8.9) | 121 (11.3) | 590 (13.2) | 40 (5.8) |

women, and physical activity in women. The results of the mediation analysis using the AFT model are provided in Table 4. In women, low SES on average was associated with an 18% shorter survival time until their first CVD compared to medium SES. Of this effect, 1% could be attributed to depressed mood (NIE 0.99, 95% CI 0.97–0.999) and 2% to perceived anxiety/depression (NIE 0.98, 95% CI 0.97–0.99). However, none of the results were statistically significant in men.

**Table 4. Adjusted direct and indirect associations of socioeconomic status with cardiovascular disease via potential intermediary variables.**

| Mediator | SES | Natural direct effect | | Natural indirect effect | | Total effect | | PM |
|---|---|---|---|---|---|---|---|---|
| | | Estimate | 95% CI | Estimate | 95% CI | Estimate | 95% CI | |
| **Men** | | | | | | | | |
| Depressed mood | | | | | | | | |
| | Low | 1.03 | 0.88, 1.21 | 1.00 | 0.98, 1.01 | 1.03 | 0.88, 1.21 | −3.1% |
| | Medium | 1.00 (ref) | | 1.00 (ref) | | 1.00 (ref) | | |
| | High | 1.02 | 0.87, 1.21 | 1.00 | 1.00, 1.00 | 1.02 | 0.87, 1.21 | 2.6% |
| Perceived anxiety/depression | | | | | | | | |
| | Low | 1.04 | 0.89, 1.21 | 0.99 | 0.98, 1.00 | 1.03 | 0.88, 1.20 | −27.7% |
| | Medium | 1.00 (ref) | | 1.00 (ref) | | 1.00 (ref) | | |
| | High | 1.02 | 0.87, 1.21 | 1.00 | 1.00, 1.00 | 1.02 | 0.87, 1.21 | 4.4% |
| Smoking status | | | | | | | | |
| | Low | 1.03 | 0.89, 1.21 | 1.00 | 0.99, 1.00 | 1.03 | 0.88, 1.20 | −15.4% |
| | Medium | 1.00 (ref) | | 1.00 (ref) | | 1.00 (ref) | | |
| | High | 1.02 | 0.87, 1.21 | 1.00 | 0.99, 1.01 | 1.02 | 0.87, 1.21 | 4.5% |
| Physical activity | | | | | | | | |
| | Low | 1.05 | 0.90, 1.23 | 0.98 | 0.96, 1.00 | 1.03 | 0.88, 1.20 | −74.0% |
| | Medium | 1.00 (ref) | | 1.00 (ref) | | 1.00 (ref) | | |
| | High | 1.02 | 0.86, 1.20 | 1.01 | 1.00, 1.01 | 1.02 | 0.87, 1.21 | 23.7% |
| **Women** | | | | | | | | |
| Depressed mood | | | | | | | | |
| | Low | 0.83 ** | 0.72, 0.96 | 0.99 * | 0.97, 0.999 | 0.82 ** | 0.71, 0.95 | 6.7% |
| | Medium | 1.00 (ref) | | 1.00 (ref) | | 1.00 (ref) | | |
| | High | 1.12 | 0.77, 1.62 | 1.00 | 1.00, 1.01 | 1.12 | 0.77, 1.62 | 3.8% |
| Perceived anxiety/depression | | | | | | | | |
| | Low | 0.84 * | 0.72, 0.97 | 0.98** | 0.97, 0.99 | 0.82 ** | 0.71, 0.95 | 8.2% |
| | Medium | 1.00 (ref) | | 1.00 (ref) | | 1.00 (ref) | | |
| | High | 1.12 | 0.77, 1.62 | 1.00 | 0.99, 1.00 | 1.11 | 0.77, 1.61 | −3.3% |
| Smoking status | | | | | | | | |
| | Low | 0.82 ** | 0.71, 0.95 | 0.99 | 0.99, 1.00 | 0.82 ** | 0.71, 0.95 | 2.6% |
| | Medium | 1.00 (ref) | | 1.00 (ref) | | 1.00 (ref) | | |
| | High | 1.12 | 0.78, 1.63 | 1.00 | 1.00, 1.01 | 1.13 | 0.78, 1.63 | 2.2% |
| Physical activity | | | | | | | | |
| | Low | 0.82 ** | 0.71, 0.95 | 0.99 | 0.98, 1.01 | 0.82 ** | 0.71, 0.95 | 3.1% |
| | Medium | 1.00 (ref) | | 1.00 (ref) | | 1.00 (ref) | | |
| | High | 1.13 | 0.78, 1.63 | 1.00 | 1.00, 1.00 | 1.13 | 0.78, 1.64 | 0.4% |

* p<.05. ** p<.01. ***p<.001.

Abbreviation: SES, socioeconomic status; PM, proportion mediated; CI, confidence interval.

Note: The model was adjusted by age, residence, marital status, obesity, previous diabetes mellitus, and previous hypertension.

S4 Table presents the range of possible confounder–mediator and confounder–outcome risk ratios for the additive model in low SES women with depressed mood as a mediator. However, none of the confounding scenarios examined were consistent with a proportion explained by the adjusted PIE effect of 6.7%, indicating that the observed indirect effect is unlikely to be fully explained by unmeasured confounding. S3–S8 Figs show the impact of each confounder with a given prevalence (e.g., 5%, 20%, 40%, 60%, 80%, and 95%) in the additive model, where the PIE accounted for 2.0% of the total effect.

In sensitivity analyses using individual SES indicators, income showed similar results to the LCA-based composite SES. Low income was associated with an 15% shorter survival time than medium income, of which 2% was attributable to depressed mood and perceived anxiety/depression in women (S5 Table). In the analysis of working status, men with no employment showed a 15% significantly shorter CVD survival time compared to those with part-time employment, of which 1% was attributable to perceived anxiety/depression (S6 Table). For education, no significant associations of SES-CVD were observed in either men or women (S7 Table). For health insurance, women with NHIS showed a 27% longer CVD survival time compared to those with medical aid, of which 3% was mediated by depressed mood (S8 Table).

## Discussion

This study investigated whether psychological (depressed mood, perceived anxiety/depression) and health behavioral factors (smoking status, physical activity) mediate the association between SES and CVD incidence, using data from the KHPS. SES was categorized into low, medium, and high levels using LCA in both men and women. Low SES was associated with higher CVD risk only among women. This association was partially mediated by depressed mood, accounting for 6.7% of the total effect, and by perceived anxiety/depression, accounting for 8.2%.

This study used LCA based on household income, education, work status, and health insurance to classify SES into low, medium, and high, providing a more detailed profile than traditional indicators. Among men, those in the high SES group were typically classified as having full-time working status, whereas among women, the high SES group was characterized by having no working status. This suggests sex-specific differences in the composition of SES: among women, even those in higher SES groups may rely less on personal earnings from labor, with income instead derived from household or external resources.

A significant direct effect of low SES on increased CVD risk was observed among women, while no significant effects were found among men. Several theories have been proposed to explain why the SES–CVD association is more pronounced in women than in men. First, sex differences in the SES–CVD association may reflect unequal access to preventive care and lower risk awareness among women, leading to delayed diagnosis or treatment [22,44]. Second, women with lower education are more likely to experience social or familial circumstances, such as single parenting, that elevate cardiovascular risk [45]. Third, women with lower SES tend to experience menopause earlier, which may alter body composition and prolong exposure to CVD risk [46,47].

In the mediation analysis using the AFT model, psychological factors (depressed mood 6.7%, perceived anxiety/depression 8.2%) were found to mediate the association with low SES. These findings are consistent with previous studies that identified mental health as a mediating factor in the SES–CVD relationship [19,48]. In the study by Hua et al., depression measured by the PHQ-9 was reported to mediate 10.92% of the SES–CVD relationship, which is comparable to the findings of our study. Jones et al. found that although both anxiety and depression were associated with CVD, only depression served as a causal mediator, accounting for 2%. Socioeconomic inequality may contribute to disparities in psychological development by affecting exposure to stressors and resources for coping, which can lead to increase CVD risk through mechanisms such as chronic inflammation and heightened sympathetic nervous system activation [49–51].

Unlike previous studies, our study identified psychological factors as significant mediators among women, whereas no such effects were observed in men. A significant interaction between sex and household income on major depressive

episodes, indicating a stronger association among women [52]. Women may be more susceptible than men to depression from environmental factors such as low SES, which psychosocial gender role-related factors and sex-based differences in endocrine stress responses may explain [53,54]. Depression is a risk factor for CVD, and its impact on CVD risk is greater in women than in men [55]. In women, lifetime changes in sex hormones such as estrogen and progesterone significantly influence depression and cardiovascular health by modulating inflammation through the inhibition of pro-inflammatory cytokines and the promotion of anti-inflammatory cytokines [56,57].

We found that the each SES factors constituting the LCA exhibited different patterns between men and women. Among women, low income was associated with a reduced survival time to CVD, with depressed mood and anxiety acting as mediators. In women, the interaction between household income and major depressive episodes was found to be stronger than in men [52]. Among men, absence of working status was related to reduced survival time to CVD, with anxiety serving as a significant mediator. In men, employment had a greater impact on mental health compared to women, with having a job serving as a protective factor against depressive and anxiety disorders [58].

Our study has several strengths. First, we applied overall SES using LCA, providing a detailed SES profile that each traditional SES indicator could not capture. This approach mitigates causal-direction biases caused by using individual SES measures. Second, we employed a counterfactual causal mediation with the AFT model, which offers robust causal insights and can accommodate exposure-mediator interactions. The AFT model is preferred over the Cox model in mediation analysis, as it does not require the rare outcome assumption [39]. Third, longitudinal study offers better insights into the causal relationship than cross-sectional studies.

Despite the strengths noted above, our studies have several limitations. First, all variables were self-reported, which can lead to misclassification bias due to memory loss. In addition, psychological factors were assessed through self-reported feelings rather than physician diagnoses; therefore, future mediation analyses using clinically diagnosed factors are warranted. Second, the available dataset only covered the years 2008–2018, which may not fully reflect more recent socioeconomic and cardiovascular changes. Third, the assumptions for causal mediation analysis require no unmeasured confounding; however, factors such as social support, genetics, and environment may serve as potential unmeasured confounders. Accordingly, we conducted a sensitivity analysis; however, because the estimates were based solely on relative risk, the consideration of time was insufficient. Fourth, this study is based on Korean data, and caution should be exercised when extrapolating the findings to other contexts, such as low- and middle-income countries and countries with differing health systems and access to healthcare. In addition, the SES latent class definitions employed in this study are culturally specific and may not be directly generalizable to other populations. Lastly, when multiple mediators are involved, there is a potential for post-randomization confounders affecting the effects of other mediators, requiring caution in interpretation. Our study uses one model for each mediator, but further studies using a multiple-mediator model should be considered.

This study examined the relationship between SES and CVD incidence and explored the indirect effects of psychological factors and health behavior. We observed a sex-specific difference in the association between SES and CVD, with depressed mood and perceived anxiety/depression serving as a partial mediator among women. This suggests that differences in the distribution of psychological factors across SES subgroups contribute to CVD risk, providing insight into the mechanisms underlying socioeconomic inequalities in CVD. Our findings underscore the broader importance of considering psychological pathways in cardiovascular prevention strategies among low-SES women.

## Supporting information

**S1 Fig. Log–log plot of estimated survivor functions by socioeconomic status group in men.** Log of the negative log of the estimated survivor function plotted against the log of follow-up duration for men, stratified by SES group (three categories).
(TIFF)

**S2 Fig. Log–log plot of estimated survivor functions by socioeconomic status group in women.** Log of the negative log of the estimated survivor function plotted against the log of follow-up duration for women, stratified by SES group (three categories).
(TIFF)

**S3 Fig. Impact of an unmeasured confounder (5% prevalence) on the pure indirect effect in the additive interaction model among low socioeconomic status women, with depressed mood as the mediator.** Each line depicts the adjusted PIE that aligns with the observed PIE under specific confounder–outcome and confounder–mediator ($RR_{CM}$) risk ratios.
(TIFF)

**S4 Fig. Impact of an unmeasured confounder (20% prevalence) on the pure indirect effect in the additive interaction model among low socioeconomic status women, with depressed mood as the mediator.** Each line depicts the adjusted PIE that aligns with the observed PIE under specific confounder–outcome and confounder–mediator ($RR_{CM}$) risk ratios.
(TIFF)

**S5 Fig. Impact of an unmeasured confounder (40% prevalence) on the pure indirect effect in the additive interaction model among low socioeconomic status women, with depressed mood as the mediator.** Each line depicts the adjusted PIE that aligns with the observed PIE under specific confounder–outcome and confounder–mediator ($RR_{CM}$) risk ratios.
(TIFF)

**S6 Fig. Impact of an unmeasured confounder (60% prevalence) on the pure indirect effect in the additive interaction model among low socioeconomic status women, with depressed mood as the mediator.** Each line depicts the adjusted PIE that aligns with the observed PIE under specific confounder–outcome and confounder–mediator ($RR_{CM}$) risk ratios.
(TIFF)

**S7 Fig. Impact of an unmeasured confounder (80% prevalence) on the pure indirect effect in the additive interaction model among low socioeconomic status women, with depressed mood as the mediator.** Each line depicts the adjusted PIE that aligns with the observed PIE under specific confounder–outcome and confounder–mediator ($RR_{CM}$) risk ratios.
(TIFF)

**S8 Fig. Impact of an unmeasured confounder (95% prevalence) on the pure indirect effect in the additive interaction model among low socioeconomic status women, with depressed mood as the mediator.** Each line depicts the adjusted PIE that aligns with the observed PIE under specific confounder–outcome and confounder–mediator ($RR_{CM}$) risk ratios.
(TIFF)

**S1 Table. Summary of model fit statistics for different numbers of latent classes by sex.** Abbreviation: DF, degree of freedom; AIC, Akaike information criterion; BIC, Bayesian information criterion; ABIC, Adjusted BIC.
(DOCX)

**S2 Table. Goodness of fit of the exponential and Weibull distribution.** Abbreviations: AIC, Akaike information criterion.
(DOCX)

**S3 Table. Effect modification on multiplicative and additive scales between socioeconomic status and binary mediators by sex.** * p < .05. ** p < .01. ***p < .001. Note: The model was adjusted for age, residence, marital status,

obesity, previous diabetes mellitus, and previous hypertension. The measure of effect modification on the multiplicative scale is expressed as a time ratio (TR), derived by exponentiating the regression coefficients. The measure of effect modification on the additive scale was assessed using the relative excess risk due to interaction (RERI), calculated by converting TR into hazard ratios (HR) based on the parameter scale, and interpreting HR as an approximation of risk ratios (RR). Confidence intervals were estimated using the delta method.
(DOCX)

**S4 Table. Sensitivity analysis for unmeasured confounder in low socioeconomic status women with depressed mood as a mediator.** [a]No confounder–outcome risk ratio exists for the given confounder prevalence and confounder–mediator risk ratio.
(DOCX)

**S5 Table. Adjusted direct and indirect associations of income as an individual socioeconomic status indicator with cardiovascular disease via potential mediators.** * $p < .05$. ** $p < .01$. ***$p < .001$. Abbreviation: PM, proportion mediated; CI, confidence interval. Note: The model was adjusted by age, residence, marital status, obesity, previous diabetes mellitus, and previous hypertension. Income was categorized as low (1st quintile), middle (2nd–4th quintiles), and high (5th quintile).
(DOCX)

**S6 Table. Adjusted direct and indirect associations of working status as an individual socioeconomic status indicator with cardiovascular disease via potential mediators.** * $p < .05$. ** $p < .01$. ***$p < .001$. Abbreviation: PM, proportion mediated; CI, confidence interval. Note: The model was adjusted by age, residence, marital status, obesity, previous diabetes mellitus, and previous hypertension.
(DOCX)

**S7 Table. Adjusted direct and indirect associations of education as an individual socioeconomic status indicator with cardiovascular disease via potential mediators.** * $p < .05$. ** $p < .01$. ***$p < .001$. Abbreviation: PM, proportion mediated; CI, confidence interval. Note: The model was adjusted by age, residence, marital status, obesity, previous diabetes mellitus, and previous hypertension.
(DOCX)

**S8 Table. Adjusted direct and indirect associations of insurance as an individual socioeconomic status indicator with cardiovascular disease via potential mediators.** * $p < .05$. ** $p < .01$. ***$p < .001$. Abbreviation: PM, proportion mediated; CI, confidence interval. Note: The model was adjusted by age, residence, marital status, obesity, previous diabetes mellitus, and previous hypertension.
(DOCX)

## Author contributions

**Conceptualization:** Jiwon Choi, Sung-il Cho.

**Data curation:** Jiwon Choi.

**Methodology:** Jiwon Choi, Sung-il Cho.

**Project administration:** Sung-il Cho.

**Resources:** Jiwon Choi.

**Supervision:** Sung-il Cho.

**Validation:** Sung-il Cho.

**Visualization:** Jiwon Choi, Sung-il Cho.

**Writing – original draft:** Jiwon Choi.

**Writing – review & editing:** Jiwon Choi.

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
