## [Decision Letter · Decision Letter 0]

27 Oct 2025

Dear Dr. Cho,

We look forward to receiving your revised manuscript.

Kind regards,

Petri Böckerman

Academic Editor

PLOS ONE

Journal Requirements:

3. Thank you for uploading your study's underlying data set. Unfortunately, the repository you have noted in your Data Availability statement does not qualify as an acceptable data repository according to PLOS's standards.

At this time, please upload the minimal data set necessary to replicate your study's findings to a stable, public repository (such as figshare or Dryad) and provide us with the relevant URLs, DOIs, or accession numbers that may be used to access these data. For a list of recommended repositories and additional information on PLOS standards for data deposition, please see https://journals.plos.org/plosone/s/recommended-repositories ..

Additional Editor Comments:

The revised paper should address all comments.

Reviewers' comments:

Reviewer's Responses to Questions

**Comments to the Author**

1. Is the manuscript technically sound, and do the data support the conclusions?

Reviewer #1: Yes

Reviewer #2: Partly

Reviewer #3: Yes

2. Has the statistical analysis been performed appropriately and rigorously?

Reviewer #1: Yes

Reviewer #2: I Don't Know

Reviewer #3: Yes

3. Have the authors made all data underlying the findings in their manuscript fully available?

Reviewer #1: Yes

Reviewer #2: No

Reviewer #3: No

4. Is the manuscript presented in an intelligible fashion and written in standard English?

Reviewer #1: Yes

Reviewer #2: No

Reviewer #3: Yes

Reviewer #1: The study is methodologically solid and well executed. Minor improvements such as consistent abbreviation use, correcting typographical errors, and expanding policy recommendations; would further strengthen it.

Overall, it offers valuable insights into socioeconomic inequalities in cardiovascular health.

Reviewer #2: It seems that the study conducted is not well described and has many ambiguities.

The type of the current study is not stated.

The data related to the years 2008 to 2018, which may have resulted in the loss of valuable data due to rapid global changes.

Also, the introduction of the study cannot fully explain the necessity and importance of the study.

The research method section seems to have many shortcomings. The method of data collection should be fully explained and in some cases requires references.

How was the health behavior variable assessed? This variable has a measurable scale. But how was it collected in this study?

Also, the study discussion and conclusion section is poorly written. What are the implications of the present results for health promotion in society? What are the basic and practical suggestions for health policymakers?

Literary review supports the study variables, especially the mediating variables, in terms of their effect on the main dependent variable, which is cardiovascular disease, not being well expressed.

This study also has many methodological ambiguities. Did a statistician oversee the data and how the study was analyzed?

The structure of the present manuscript is not fluent and does not create interest in conveying the research results.

It seems that if the present study had been conducted in a smaller sample population but in a more precise manner and the variables, especially the health behavior variable, had been examined with an appropriate scale, it would have had higher scientific value.

Reviewer #3: This manuscript investigates the association between socioeconomic status (SES) and cardiovascular disease (CVD) incidence by sex, exploring mediation by health behaviors, depression, and unmet medical needs using data from the Korea Health Panel Survey (KHPS) 2009–2018. Employing latent class analysis (LCA) to derive SES and causal mediation analysis (AFT model), the authors find that low SES is significantly associated with shorter survival to CVD among women, partially mediated by depression.

Here are the main points that should be addressed before publication:

• The manuscript must clearly articulate the timing of measurement for SES, mediators, and CVD incidence and discuss whether causal mediation assumptions (such as no unmeasured confounding between mediator and outcome) are plausible. Without temporal separation, the mediation interpretation becomes weaker.

• How is depression measured (e.g., PHQ-9 score, clinical diagnosis)? The validity and cut‐off used need clear description.

• Unmet medical needs: This is often self-reported and prone to recall/reporting bias. The manuscript should discuss how this variable is operationalized and its reliability.

• Cardiovascular disease (CVD): Is this a self-reported physician diagnosis, hospital record, or self-report? Are incident vs prevalent cases distinguished? Clarity is needed.

• The authors did not perform a priori power or sample-size calculation. They simply used all available respondents (n = 11 397) from the KHPS dataset without stating the detectable effect size, statistical power, or design effect.

• The dataset shows approximately 79% rural residents, much higher than Korea’s real rural proportion (18 %), implying sampling distortion. It needs clarification.

• The manuscript must discuss potential confounders (e.g., comorbidities, genetic risk, baseline health status) and what has been adjusted for. If some confounders are unmeasured, this weakens the ability to interpret the mediation effects causally.

• The study is based on Korean data; this should be noted in limitations regarding extrapolation to other contexts (e.g., LMICs, differing health systems, and access issues). Also, the SES latent class definitions will be culturally specific.

• The use of abbreviations (e.g., SES and TE) should be clearly explained at first mention in each section.

**Do you want your identity to be public for this peer review?** For information about this choice, including consent withdrawal, please see our For information about this choice, including consent withdrawal, please see our Privacy Policy .

Reviewer #1: **Yes:** Faisal AlkulaibFaisal Alkulaib

Reviewer #2: No

Reviewer #3: No

---

## [Author Response · Author response to Decision Letter 1]

21 Dec 2025

5. Review Comments to the Author

Reviewer #1: The study is methodologically solid and well executed. Minor improvements such as consistent abbreviation use, correcting typographical errors, and expanding policy recommendations; would further strengthen it.

Overall, it offers valuable insights into socioeconomic inequalities in cardiovascular health.

We deeply appreciate to carefully reading our text. We have now carefully considered the reviewer’s comments and reflected them in the revised manuscript to improve clarity and precision.

● In the revised manuscript, we have ensured that all abbreviations, including socioeconomic status (SES) and total effect (TE), are clearly spelled out at their first appearance in each section for clarity and readability.

[Page 2 Lines 12-14] Previous research shows that low socioeconomic status (SES) increases the risk of cardiovascular disease (CVD) and contributes to health disparities through the unequal distribution of intermediary factors.

[Page 16 Lines 238-243] In women, low SES on average had a 17% shorter survival time until their first CVD compared to medium SES, and 2% could be attributed to depressed mood (NDE 0.83, 95% confidence interval (CI) 0.72-0.97; NIE 0.98, 95% CI 0.97-0.999; Total effect (TE) 0.82, 95% CI 0.71-0.95) and perceived anxiety/depression (NDE 0.84, 95% CI 0.73-0.97; NIE 0.98, 95% CI 0.97-0.99; TE 0.83, 95% CI 0.72-0.96).

[Page 18 Line 247] Abbreviation: SES, socioeconomic status; PM, proportion mediated; CI, confidence interval.

● We have corrected typographical errors and grammatically reviewed the entire text.

● We have added specific policy recommendations to the discussion section.

[Page 22 Lines 343-349] Therefore, policies such as community‑based mental health screening and accessible treatment services for low‑SES women should be implemented to reduce CVD risk. Targeted interventions should prioritize women in low‑SES groups by integrating subsidized mental health services into primary care. In addition, coordinated policies that combine socioeconomic support with accessible psychological care for low‑SES women can reduce health inequities and ultimately lower CVD risk. Our findings underscore the broader importance of integrating mental health into cardiovascular prevention strategies to reduce CVD risk among low‑SES women.

Reviewer #2: It seems that the study conducted is not well described and has many ambiguities.

We appreciate you taking the time to read our text attentively. We have carefully taken into account the reviewer's comments and included them in the revised manuscript to improve the novelty of our study.

• The type of the current study is not stated.

Thank you for the valuable comment. We have added the study type to the Materials and Methods section.

[Page 6 Lines 93-95] This retrospective cohort study utilized the Korea Health Panel Survey (KHPS) data (version 1.7.3.), which was conducted annually from 2008 to 2018 by the Korea Institute for Health and Social Affairs and the National Health Insurance Service (NHIS) consortium.

• The data related to the years 2008 to 2018, which may have resulted in the loss of valuable data due to rapid global changes.

We appreciate the reviewer’s insightful comment regarding the study period. Unfortunately, the available dataset only covered the years 2008 to 2018, and therefore we were unable to include more recent data in our analysis. We acknowledge that rapid global changes after 2018 may influence the outcomes; however, our study was designed to utilize the most comprehensive data accessible at the time. Despite this limitation, the large sample size and robust methodology provide meaningful and reliable findings that contribute to the understanding of the research question. We have added a statement in the limitations section to clarify this point.

[Page 22 Lines 325-327] Second, the available dataset only covered the years 2008 to 2018, which may not fully reflect more recent socioeconomic and cardiovascular changes.

• Also, the introduction of the study cannot fully explain the necessity and importance of the study.

Thank you for your precious comment. To emphasize the significance of this study, we revised the title and thoroughly rewrote the introduction as follows:

Title: Association between socioeconomic status and cardiovascular disease by sex: mediating roles of psychological and behavioral factors

• The first paragraph presents the current status of cardiovascular disease (CVD).

• The second paragraph discusses CVD risk factors, highlights the association with socioeconomic status (SES), and introduces the conceptual framework of mediating factors.

• The third paragraph examines the mechanisms of psychological factors such as depression and anxiety in the relationship between low SES and CVD.

• The fourth paragraph addresses multifaceted social factors and health behaviors as mediating mechanisms, explaining smoking and physical activity as pathways to CVD.

• The fifth paragraph reviews previous studies on mediators and identifies their limitations.

• The sixth paragraph points out the shortcomings of using single SES indicators and emphasizes the need for composite measures, while the seventh paragraph provides examples of SES research employing latent class analysis (LCA).

• The final paragraph highlights the gaps in the existing literature and states the aim of the present study.

• The research method section seems to have many shortcomings. The method of data collection should be fully explained and in some cases requires references.

We provided a more detailed description of data collection in the manuscript and additionally included a reference to a study that explains data collection from the Korea Health Panel.

[Page 6 Lines 95-101] This panel employs a multi-stage stratified probability sampling method using 90% of the 2005 Population and Housing Census data across sixteen districts nationwide to maintain national representativeness. KHPS collected variables through face-to-face interviews using the Computer Assisted Personal Interviewing (CAPI) method [29]. Although the data were self-reported, their reliability was enhanced by the panel households’ completion of a one-year health diary on medical utilization and the collection of medical receipts and year-end tax settlement records.

[Page 7 Lines 124-127] The main outcome was newly diagnosed CVD cases, assessed through self reported physician diagnoses. Participants reported the physician’s diagnosis name and the corresponding International Classification of Diseases 10th edition (ICD-10) code during surveys of emergency, inpatient, and outpatient service use.

*reference: Jin, D. L., Go, D. S., & Yoon, S. J. (2025). Longitudinal perspectives on health and medical research in Korea: strengths and limitations of key panel datasets. Journal of Korean Medical Science, 40(23).

We also supplemented the statistical explanation of the mediation analysis.

[Page 9 Lines 167-177] Causal mediation analysis provides a more general and rigorous framework for survival models, accommodating nonlinear relationships and interactions [34]. It needs assumptions for valid estimation of natural direct effects (NDE) and natural indirect effects (NIE): measured covariates must control for confounding in the (1) exposure–outcome, (2) mediator–outcome, and (3) exposure–mediator relationships, and (4) mediator–outcome confounders must not be influenced by the exposure [35]. The accelerated failure time (AFT) model was formulated using survival functions, with the Weibull distribution showing the best fit based on AIC criteria (S2 Table). The AFT model represents time ratios, interpreted as a reduction (less than 1) for deleterious covariates and an increase (greater than 1) for protective covariates. The proportion mediated (PM) was calculated following the method proposed by VanderWeele and Vansteelandt [36]. Mediation analysis was conducted using a SAS macro that incorporates causal effects estimated on the mean survival ratio scale within the AFT model [37].

• How was the health behavior variable assessed? This variable has a measurable scale. But how was it collected in this study?

In Korea, health behavior scales include measures such as alcohol consumption, stress and mental health, physical activity, etc (Shin, 2010). However, previous meta-analyses have shown that alcohol consumption of approximately 2.5–14.9 g/day (about ≤1 drink per day) may be associated with a reduced risk of CVD (Ronksley et al, 2011). Therefore, examining a composite scale that includes alcohol intake may not be appropriate when evaluating its association with CVD; rather, it may be more suitable to assess health behaviors as independent indicators.

Unfortunately, the survey instrument used to collect data in the present study does not include a health behavior scale, making it difficult to adopt such a scale directly. Instead, based on a review of the existing literature, we identified behavioral factors that mediate the relationship between SES and CVD. We found that in studies examining the association between socioeconomic position and mortality from cardiovascular diseases, behavioral mediators were defined as smoking, alcohol consumption, physical inactivity, poor diet, and body mass index (Hossin et al, 2021). Therefore, in this study, health behaviors were defined using two variables: physical activity and smoking status.

*references:

Shin, Y. H. (2010). Development and psychometric evaluation of a scale to measure health behaviors of adolescents. Journal of Korean Academy of Nursing, 40(6), 820-830.

Ronksley, P. E., Brien, S. E., Turner, B. J., Mukamal, K. J., & Ghali, W. A. (2011). Association of alcohol consumption with selected cardiovascular disease outcomes: a systematic review and meta-analysis. Bmj, 342, d671.

Hossin, M. Z., Koupil, I., & Falkstedt, D. (2021). Early life socioeconomic position and mortality from cardiovascular diseases: an application of causal mediation analysis in the Stockholm Public Health Cohort. BMJ open, 9(6), e026258.

• Also, the study discussion and conclusion section is poorly written. What are the implications of the present results for health promotion in society? What are the basic and practical suggestions for health policymakers?

We appreciate the reviewer’s comment regarding the discussion and conclusion section. We have added concrete recommendations, including community‑based screening, integration of subsidized mental health services into primary care, and coordinated socioeconomic support policies for low‑SES women.

[Page 22 Lines 343-349] Therefore, policies such as community‑based mental health screening and accessible treatment services for low‑SES women should be implemented to reduce CVD risk. Targeted interventions should prioritize women in low‑SES groups by integrating subsidized mental health services into primary care. In addition, coordinated policies that combine socioeconomic support with accessible psychological care for low‑SES women can reduce health inequities and ultimately lower CVD risk. Our findings underscore the broader importance of integrating mental health into cardiovascular prevention strategies to reduce CVD risk among low‑SES women.

• Literary review supports the study variables, especially the mediating variables, in terms of their effect on the main dependent variable, which is cardiovascular disease, not being well expressed.

We appreciate the reviewer’s comment regarding the insufficient expression of mediating variables in the literature review. In revision, we have expanded the introduction to emphasize how psychological factors (depressed mood, perceived anxiety/depression) and behavioral factors (smoking, physical activity) contribute to CVD. We now highlight evidence showing that low SES is linked to higher rates of depression and perceived anxiety/depression, which elevate CVD risk through biological stress pathways, while smoking and physical inactivity further increase risk as key modifiable behaviors. We also incorporated findings from prior studies quantifying these mediating effects and noted their limitations.

[Page 3 Lines 45 - Page 4 Lines 68] Low income is associated with having depression and anxiety, both of which represent established independent risk factors for CVD [9, 10]. Meta-analyses have shown that low income is associated with increased depression [11]. Individuals with depression may experience dysregulation of the sympathetic nervous system and the hypothalamic–pituitary–adrenal axis, leading to coronary vasoconstriction, endothelial dysfunction, and increased platelet activation, ultimately elevating cardiac risk [10]. Structured diagnostic interviews suggest that lower SES is associated with a higher likelihood of anxiety [12]. Stress and anxiety can promote atherosclerosis and may serve as acute triggers of major cardiac events, thereby increasing CVD risk [13, 14].

Health behaviors are recognized as key mediating mechanisms linking distal structural factors to individual health outcomes [15]. Health behaviors are shaped by multifaceted social, economic, and environmental factors and they exhibit strong social patterning [16]. Smoking is a major risk factor for CVD, and individuals who smoke have a significantly higher risk of CVD compared to non-smokers [17]. Sedentary behavior and physical activity are modifiable risk factors for CVD, with sedentary behavior increasing CVD risk and higher levels of physical activity reducing CVD risk [18]. These results can be explained by physical activity improving lipid profiles—raising HDL and lowering triglycerides and total cholesterol—which in turn reduces CVD risk.

Previous studies have explored psychological and behavioral mediators in the SES–CVD association. Depression was reported to account for 10.9% of this relationship, although the cross-sectional design limited causal inference [19]. Other research identified smoking, physical inactivity, and alcohol use as mediators of the SES–ischemic heart disease mortality link [20], while cigarette smoking and physical inactivity explained 11.4% and 7.7% of the association between social determinants of health (SDOH) and CVD mortality, respectively [21]. However, the use of a composite SDOH score based on the number of unfavorable indicators makes it difficult to disentangle the specific contribution of each attribute.

• This study also has many methodological ambiguities. Did a statistician oversee the data and how the study was analyzed?

Thank you for your precious comment. We have provided a detailed description of the methods section, elaborated on causal mediation analysis, and added relevant references.

[Page 9 Lines 167-177] Causal mediation analysis provides a more general and rigorous framework for survival models, accommodating nonlinear relationships and interactions [34]. It needs assumptions for valid estimation of natural direct effects (NDE) and natural indirect effects (NIE): measured covariates must control for confounding in the (1) exposure–outcome, (2) mediator–outcome, and (3) exposure–mediator relationships, and (4) mediator–outcome confounders must not be influenced by the exposure [35]. The accelerated failure time (AFT) model was formulated using survival functions, with the Weibull distribution showing the best fit based on AIC criteria (S2 Table). The AFT model represents time ratios, interpreted as a reduction (less than 1) for deleterious covariates and an increase (greater than 1) for protective covariates. The proportion mediated (PM) was calculated following the method proposed by VanderWeele and Vansteelandt [36]. Mediation analysis was conducted using a SAS macro that incorporates causal effects estimated on the mean survival ratio scale within the AFT model [37].

• The structure of the present manuscript is not fluent and does not create interest in conveying the research results.

We thank the revi

---

## [Decision Letter · Decision Letter 1]

6 Jan 2026

Dear Dr. Cho,

Thank you for submitting your manuscript to PLOS ONE. After careful consideration, we feel that it has merit but does not fully meet PLOS ONE’s publication criteria as it currently stands. Therefore, we invite you to submit a revised version of the manuscript that addresses the points raised during the review process.

We look forward to receiving your revised manuscript.

Kind regards,

Belal Hossain, PhD

Academic Editor

PLOS One

**Journal Requirements:**

**Additional Editor Comments:**

The manuscript is well written, and the reviewers were satisfied with the revised version. However, there are some major points that need to be addressed before I consider this paper for acceptance.

1. The study conclusion should be revised (in both the Abstract and the Discussion section). The conclusion should be based on the findings of the current work rather than on hypothetical assumptions/recommendations.

2. The rationale for sex-specific estimates is thin. The authors did not justify presenting sex-based estimates.

3. It is unclear whether all analyses were conducted separately for males and females.

4. It is unclear how the 'proportion mediated' was calculated. The author cited a reference, but it was for odds ratios, whereas the present study used AFT models.

5. The authors mentioned 'interactions' by sex in many places. I assumed the authors were referring to 'effect modification'. In either case, the recommendations for presenting analyses of effect modification/interaction were not followed.

6. There should be at least one sensitivity analysis using a Cox proportional hazards model, as it is most frequently used for modeling survival outcomes.

7. The authors cited a reference, but I strongly recommend describing which mediation analysis technique was used, such as the counterfactual framework-based weighting approach.

Reviewers' comments:

Reviewer's Responses to Questions

**Comments to the Author**

Reviewer #1: All comments have been addressed

Reviewer #3: All comments have been addressed

2. Is the manuscript technically sound, and do the data support the conclusions?

Reviewer #1: Yes

Reviewer #3: Yes

3. Has the statistical analysis been performed appropriately and rigorously?

Reviewer #1: Yes

Reviewer #3: Yes

4. Have the authors made all data underlying the findings in their manuscript fully available?

Reviewer #1: Yes

Reviewer #3: No

5. Is the manuscript presented in an intelligible fashion and written in standard English?

Reviewer #1: Yes

Reviewer #3: Yes

Reviewer #1: The authors have satisfactorily addressed all comments from the previous review. The revised manuscript is clearer and methodologically sound, with improved description of study design, mediator definitions, and causal mediation assumptions.

The statistical analyses are appropriate and sufficiently detailed, and the added sensitivity analyses support the robustness of the findings. The discussion has been strengthened, particularly in interpreting sex-specific results and policy implications, and the limitations are adequately acknowledged.

Overall, the manuscript meets the technical and scientific standards of PLOS ONE and is acceptable for publication in its current form.

Reviewer #3: I would like to inform you that all reviewer comments have been fully addressed. The manuscript has been revised accordingly and is now ready for publication.

**Do you want your identity to be public for this peer review?** For information about this choice, including consent withdrawal, please see our For information about this choice, including consent withdrawal, please see our Privacy Policy .

Reviewer #1: **Yes:** Faisal AlkulaibFaisal Alkulaib

Reviewer #3: No

---

## [Author Response · Author response to Decision Letter 2]

28 Feb 2026

Journal Requirements:

In accordance with the editor’s comments, we reviewed and added literature showing that the strength of the association between socioeconomic status (SES) and cardiovascular disease (CVD) differs by sex, and that mediating factors such as smoking and psychological influences may operate differently in men and women. We also examined the literature on recommendations for presenting analyses of effect modification/interaction, and incorporated values for effect modification on both the additive and multiplicative scales into the manuscript.

Additional Editor Comments:

The manuscript is well written, and the reviewers were satisfied with the revised version. However, there are some major points that need to be addressed before I consider this paper for acceptance.

We deeply appreciate to carefully reading our text. We have now carefully considered the editor’s comments and reflected them in the revised manuscript to improve precision.

1. The study conclusion should be revised (in both the Abstract and the Discussion section). The conclusion should be based on the findings of the current work rather than on hypothetical assumptions/recommendations.

Thank you for your valuable comment. We have revised the Abstract and Discussion conclusions to focus strictly on our findings. The revised conclusion now avoids hypothetical recommendations and emphasizes the interpretive significance of our results.

[Page 2 Lines 28-31] Our findings indicate that psychological factors partially mediate the association between low SES and CVD among women, highlighting sex‑specific pathways in socioeconomic health disparities and underscoring the importance of incorporating mental health considerations into cardiovascular prevention strategies.

[Page 21 Lines 372-378] This study examined the relationship between SES and CVD incidence and explored the indirect effects of psychological factors and health behavior. We observed a sex-specific difference in the association between SES and CVD, with depressed mood and perceived anxiety/depression serving as a partial mediator among women. This suggests that differences in the distribution of psychological factors across SES subgroups contribute to CVD risk, providing insight into the mechanisms underlying socioeconomic inequalities in CVD. Our findings underscore the broader importance of considering psychological pathways in cardiovascular prevention strategies among low SES women.

2. The rationale for sex-specific estimates is thin. The authors did not justify presenting sex-based estimates.

In the revised manuscript, we have summarized previous studies to highlight that analyses conducted separately by sex demonstrate differences in the influence of mediating factors between men and women. In addition, we have incorporated a new paragraph reviewing the literature showing that the strength of the SES–CVD association varies by sex, and that mediating factors such as smoking and psychological influences may operate differently across men and women.

[Page 4 Lines 65-72] Previous studies have explored psychological and behavioral mediators in the SES–CVD association. Depression was reported to account for 10.9% of this relationship, although the cross-sectional design limited causal inference [19]. Cigarette smoking and physical inactivity explained part of the association between SDOH and CVD mortality, but reliance on a composite SDOH score obscured the specific contributions of individual attributes [20]. Other research has identified smoking, physical inactivity, alcohol use, and body mass index (BMI) as significant mediators of the association between SES and ischemic heart disease mortality in both men and women; however, the proportions mediated differed by sex [21].

[Page 4 Lines 73-84] The association between SES and CVD, as well as the mediating factors, appears to differ between men and women. Low SES has been linked to higher CVD risk, with this trend more pronounced in women [22, 23]. This is because women with lower SES tend to exhibit a worse profile of cardiovascular biomarkers [24]; however, the influence of SES on factors such as diet, physical activity, and psychosocial aspects may also differ by sex [22]. Importantly, the pathways through which SES influences CVD may differ in strength between men and women. For example, smoking substantially contributes to socioeconomic inequalities in ischemic heart disease mortality, accounting for a larger proportion among men (29%) than women (16%) [21]. In contrast, psychosocial factors—particularly depression and stress—along with behavioral and lifestyle influences, disproportionately affect women, with depression being approximately twice as prevalent in women as in men, thereby increasing their risk of CVD [25]. Therefore, these observations underscore the need for sex-specific mediation analyses to better understand the underlying pathways.

3. It is unclear whether all analyses were conducted separately for males and females.

As the editor pointed out, the sex-specific analyses in this study are not clearly presented. In the LCA, SES was derived from the overall population and then stratified by sex, after which mediation analyses were conducted separately for men and women. In this study, the sample consisted of 5,034 men and 6,231 women. When LCA was applied to the overall population (original version), men were classified as 12.6% low SES, 63.2% medium SES, and 24.2% high SES, whereas women were classified as 17.5% low SES, 71.4% medium SES, and 11.0% high SES.

We subsequently examined whether the measurement of latent classes was invariant across sex (1). A chi-square difference test, based on the differences in two G² statistics and their associated degrees of freedom, indicated a p-value less than 0.05, confirming that class membership probabilities differed significantly between men and women. Therefore, because measurement invariance did not hold across gender, we conducted LCA separately for men and women.

(1) https://support.sas.com/resources/papers/proceedings16/5500-2016.pdf

We revised the overall analysis by conducting LCA separately for men and women, and specified this procedure in the Methods section. When LCA was conducted separately for men and women, high SES in women was characterized by non-working status, which differed from men. However, the analysis showed similar results, as among women, low SES was associated with a shorter average survival time until CVD, partially mediated by depressed mood and by perceived anxiety/depression.

[Page 2 Lines 20-21] SES was derived using latent class analysis based on four variables: income, education, working status, and health insurance, conducted separately by sex.

[Pages 6-7 Lines 131-133] The exposure variable was SES classified as high, medium, and low, based on LCA using four variables—self-reported household income, education level, working status, and health insurance—conducted separately for men and women.

[Page 8 Lines 176-178] We determined the number of latent classes (ranging from two to five) based on model fit statistics and theoretical interpretability in analyses conducted separately for men and women (S1 Table).

[Pages 8-9 Lines 182-184] The three-class model for both men and women demonstrated the best fit based on AIC, BIC, and G² values. Its entropy values (0.63 for men and 0.69 for women) further indicate good class interpretability and separation.

[Page 9 Lines 190-194] The SES-specific curves for both sexes were non-parallel with differing slopes, indicating a violation of the proportional hazards assumption (S1-S2 Figs). However, their approximately linear form supported the Weibull assumption, justifying the use of an AFT model. The AFT model was formulated using survival functions, with the Weibull distribution showing the best fit based on AIC criteria in both men and women (S2 Table).

[Page 12 Lines 249-258] We evaluated the characteristics of each latent class separately for men and women based on mean posterior and item-response probabilities (Table 2). In men, latent class 1 (n = 639, 12.7%) was classified as low SES, reflecting low income, low education, and non-working status. Class 2 (n = 3,177, 63.1%) represented medium SES, with intermediate levels of income and education. Class 3 (n = 1,218, 24.2%) corresponded to high SES, characterized by high income, higher education, and full-time employment. In women, latent class 1 (n = 4,471, 71.8%) represented medium SES, with intermediate income and education. Class 2 (n = 685, 11.0%) corresponded to high SES, defined by high income and higher education. Class 3 (n = 1,075, 17.3%) was classified as low SES, reflecting low income, low education, and non-working status. In all analyses, the reference category was set as medium SES, as it had the largest sample size and allowed for the assessment of trends across low and high SES groups.

[Page 12 Lines 259-260] Table 2. Mean posterior and item-response probabilities from sex-specific three-class latent class models.

[Page 13 Line 262] Table 3 presents the distribution of SES obtained from sex-specific LCA.

[Page 19 Lines 307-313] This study used LCA based on household income, education, work status, and health insurance to classify SES into low, medium, and high, providing a more detailed profile than traditional indicators. Among men, those in the high SES group were typically classified as having full-time working status, whereas among women, the high SES group was characterized by having no working status. This suggests sex-specific differences in the composition of SES: among women, even those in higher SES groups may rely less on personal earnings from labor, with income instead derived from household or external resources.

4. It is unclear how the 'proportion mediated' was calculated. The author cited a reference, but it was for odds ratios, whereas the present study used AFT models.

As the reviewer correctly pointed out, the cited reference pertained to odds ratios and was therefore inappropriate for our study. We verified the correct procedure using the macro we employed and have now revised the manuscript to include a detailed description of how the proportion mediated was calculated.

[Pages 9-10 Lines 208-211] The proportion mediated (PM) was calculated on the mean survival ratio scale, as implemented in the SAS macro for causal mediation analysis [39]. Given that the TE in an AFT model decomposes multiplicatively into the NDE and NIE (i.e., TE = NDE * times NIE), the PM was defined as [NDE * times (NIE - 1)] / (TE - 1).

5. The authors mentioned 'interactions' by sex in many places. I assumed the authors were referring to 'effect modification'. In either case, the recommendations for presenting analyses of effect modification/interaction were not followed.

We appreciate the reviewer’s insightful comment regarding the presentation of our interaction analysis. We used the term "interaction" because our analysis was based on VanderWeele’s mediation macro, which incorporates an exposure–mediator interaction term in the regression models (1, 2). In our study, we likewise included SES × mediator interaction terms in the Weibull regression models and evaluated the statistical significance of these interaction terms with p-values.

However, we agree that following the recommended guidelines for reporting effect modification and interaction enhances the clarity of our findings. Therefore, following the published recommendations by Knol and VanderWeele (2012), we evaluated the effect modification between SES and mediator variables on both additive and multiplicative scales (3). In this response letter, we provide the detailed results of these analyses, using depressed mood in the male population as a representative example. For men, the RERI (Additive Scale) for low and high SES were -0.31 (95% CI: -0.84, 0.22) and -0.42 (95% CI: -0.83, -0.01), respectively. The multiplicative TRs were 1.22 (95% CI: 0.75, 2.00) and 1.36 (95% CI: 0.53, 3.46). Accordingly, our evaluation of effect modification indicated that high SES was associated with agonistic effects on the additive scale, while no significant effects were observed on the multiplicative scale.

(1) Valeri, L., & VanderWeele, T. J. (2015). SAS macro for causal mediation analysis with survival data. Epidemiology, 26(2), e23-e24.

(2) Valeri, L., & VanderWeele, T. J. (2013). Mediation analysis allowing for exposure–mediator interactions and causal interpretation: theoretical assumptions and implementation with SAS and SPSS macros. Psychological methods, 18(2), 137.

(3) Knol, M. J., & VanderWeele, T. J. (2012). Recommendations for presenting analyses of effect modification and interaction. International journal of epidemiology, 41(2), 514-520.

Socioeconomic status (Men) Time ratio (95%CI) for low SES within strata of depressed mood Time ratio (95%CI) for high SES within strata of depressed mood

low medium high

Time ratio (95%CI) Time ratio (95%CI) Time ratio (95%CI)

Depressed mood Yes 1.14 (0.76, 1.71) 0.92 (0.71, 1.20) 1.29 (0.53, 3.13) 1.22 (0.76, 1.96) 1.39 (0.55, 3.48)

No 1.00 (0.86, 1.17) 1.0 1.02 (0.84, 1.25) 1.00 (0.86, 1.17) 1.02 (0.84, 1.25)

Time ratio (95%CI) for depressed mood within strata of SES 1.13 (0.75, 1.72) 0.93 (0.71, 1.20) 1.26 (0.51, 3.09)

Measure of interaction on additive scale (95%CI) -0.31 (-0.84, 0.22) -0.42 (-0.83, -0.01)

Measure of interaction on multiplicative scale (95%CI) 1.22 (0.75, 2.00) 1.36 (0.53, 3.46)

Notes: TRs are adjusted for age, residence, marital status, obesity, previous diabetes mellitus, and previous hypertension. The measure of effect modification on the additive scale was assessed using the Relative Excess Risk due to Interaction (RERI), calculated by converting TR into Hazard Ratios (HR) based on the parameter scale, and interpreting HR as an approximation of Risk Ratios (RR). Confidence intervals were estimated using the delta method.

We described effect modification more explicitly in the Methods and Results sections and added analyses of effects on both the multiplicative and additive scales to S3 Table.

[Page 10 Lines 213-219] For sensitivity analysis, we tested the effect modification on both the multiplicative and additive scales between SES and the binary mediators (S3 Table), in order to incorporate these findings into the causal mediation analysis. Effect modification was evaluated on the multiplicative scale using TRs derived from exponentiated regression coefficients. On the additive scale, it was assessed using the relative excess risk due to interaction (RERI), calculated by converting TRs into hazard ratios via the parameter scale and interpreting hazard ratios as an approximation of risk ratios. Confidence intervals were estimated using the delta method.

[Page 15 Lines 271-275] Multiplicative effects of mediators between SES and CVD showed no statistically significant associations in either men or women across all SES levels (S3 Table). On the additive scale, antagonistic effects were observed for high SES combined with depressed mood (in both men and women), perceived anxiety/depression in men, smoking status in women, and physical activity in women.

S3 Table. Effect modification on multiplicative and additive scales between socioeconomic status and binary mediators by sex.

Measure of effect modification on multiplicative scale

Men Women

Time ratio

(95% CI) P-value Time ratio

(95% CI) P-value

Low SES: having depressed mood 1.22

(0.75, 2.00) 0.422 1.16

(0.82, 1.65) 0.390

High SES: having depressed mood 1.36

(0.53, 3.46) 0.520 Inf

(0, Inf) 0.999

Low SES: having perceived anxiety/depression 1.01

(0.69, 1.49) 0.951 1.00

(0.74, 1.35) 0.998

High SES: having perceived anxiety/depression 1.35

(0.68, 2.67) 0.389 0.89

(0.29, 2.72) 0.845

Low SES: ever smoker 0.61

(0.36, 1.03) 0.066 0.86

(0.54, 1.37) 0.523

High SES: ever smoker 1.19

(0.79, 1.81) 0.404 Inf

---

## [Editor Report · Decision Letter 2]

9 Mar 2026

Association between socioeconomic status and cardiovascular disease by sex: mediating roles of psychological and behavioral factors

PONE-D-25-52168R2

Dear Dr. Cho,

We’re pleased to inform you that your manuscript has been judged scientifically suitable for publication and will be formally accepted for publication once it meets all outstanding technical requirements.

Kind regards,

Belal Hossain, PhD

Academic Editor

PLOS One
---

## [Editor Report · Acceptance letter]

PONE-D-25-52168R2

PLOS One

Dear Dr. Cho,

I'm pleased to inform you that your manuscript has been deemed suitable for publication in PLOS One. Congratulations! Your manuscript is now being handed over to our production team.

Kind regards,

on behalf of

Dr. Belal Hossain

Academic Editor

PLOS One